# Alternatives in Molecular Diagnostics of *Encephalitozoon* and *Enterocytozoon* Infections

**DOI:** 10.3390/jof6030114

**Published:** 2020-07-22

**Authors:** Alexandra Valenčáková, Monika Sučik

**Affiliations:** Department of Biology and Genetics, University of Veterinary Medicine and Pharmacy, Komenského 73, 04181 Košice, Slovakia; monikasucik@gmail.com

**Keywords:** *Encephalitozoon* spp., *Enterocytozoon bieneusi*, diagnosis, molecular diagnosis, primers

## Abstract

Microsporidia are obligate intracellular pathogens that are currently considered to be most directly aligned with fungi. These fungal-related microbes cause infections in every major group of animals, both vertebrate and invertebrate, and more recently, because of AIDS, they have been identified as significant opportunistic parasites in man. The Microsporidia are ubiquitous parasites in the animal kingdom but, until recently, they have maintained relative anonymity because of the specialized nature of pathology researchers. Diagnosis of microsporidia infection from stool examination is possible and has replaced biopsy as the initial diagnostic procedure in many laboratories. These staining techniques can be difficult, however, due to the small size of the spores. The specific identification of microsporidian species has classically depended on ultrastructural examination. With the cloning of the rRNA genes from the human pathogenic microsporidia it has been possible to apply polymerase chain reaction (PCR) techniques for the diagnosis of microsporidial infection at the species and genotype level. The absence of genetic techniques for manipulating microsporidia and their complicated diagnosis hampered research. This study should provide basic insights into the development of diagnostics and the pitfalls of molecular identification of these ubiquitous intracellular pathogens that can be integrated into studies aimed at treating or controlling microsporidiosis.

## 1. Introduction

Microsporidia are obligate intracellular parasites, capable of infecting almost all animal species, including humans, fish, bees and other insects. Some species of microsporidia have a very narrow host range, while others have a relatively broad host range [1]. Interestingly, host ranges can be extended when microsporidia meet immunocompromised hosts [2]. Similar to most intracellular parasites, microsporidia are highly specialized and extremely sophisticated infectious organisms. In the evolution of these organisms, adaptive mechanisms needed for their existence inside cells have been developed. This adaptation is primarily characterized as reduction [3]. In comparison with other eukaryotic cells, microsporidia are highly reduced in every aspect, from their morphology and ultrastructure to their biochemistry and metabolism, and equally on their molecular level in genes and genomes.

More than 1500 microsporidian species have been described so far, and new species are discovered on a yearly basis. Of these, *Enterocytozoon bieneusi* and *Encephalitozoon* spp., are two major genera infecting humans [4,5]. Epidemiological studies worldwide have shown that *E. bieneusi* was responsible for almost 90% of human gastrointestinal infections and *Encephalitozoon* species, including *E. intestinalis*, *E*. *cuniculi* and *E*. *hellem* are usually reported from the remaining 10% [6]. Infectious spores contaminating body excreta (urine, feces, and respiratory excreta) of animals or persons infected by microsporidia that spread into the external environment are the main sources of infection [5,7]. The horizontal transmission of the infection is most often carried out in the uro-urinary and/or the oro-fecal manner. Vertical transplacental transmission has been demonstrated in several animal species [8,9].

The developmental cycle of microsporidia involves three successive phases. During the infectious phase (spore stage), the spore enters the external environment and from there into the body of the new host. In it, the mechanism of spore germination is chemically stimulated, during which its infectious content (sporoplasm) penetrates into the host cell. The proliferative-vegetative phase takes place inside the infected host cell in two stages, the merogony stage and the schizogony stage, which aim to multiply the parasite. The result of the third phase, called intercellular sporogony, is the formation of full-fledged and infection-capable spores that reach maturity as early as 48 h after the infection [10]. Results from previous studies show the presence of microsporidia in raw wastewater, tertiary sewage water, surface and underground water, and also in recreational waters, drinking water and soil [11,12,13,14]. There is also a case of a possible outbreak of microsporidiosis from water in Lyon (France) in summer 1995 [15]. The growing trend of waterborne outbreaks associated with the agricultural production of raw materials and contaminated food, suggests that eating fresh food is not always risk free [16]. This is confirmed by the Polish author who reported incidence of infectious spores of *E. intestinalis* and *E. bieneusi* in retail foods (berries, sprouts and leafy vegetables and fruits of various kinds) of up to 11.5% [17].

Human pathogenic microsporidia are commonly considered to be pathogens with low virulence and infections caused by them are effectively regulated by the immune mechanisms, mainly by cellular immunity [18,19,20]. Microsporidia infect various types of host cells such as macrophages, histiocytes, epithelial cells (renal tubules, urinary tract, epithelial cells of the small intestine, conjunctiva and cornea, bile ducts and gall bladder, bronchi, nasal mucosa and uterus), endothelial cells of blood vessels and numerous other types of cells [21]. The difference in lesions and in tissue distribution of the parasite in humans was used as a primary criteria for the determination of different species from the genus *Encephalitozoon*—*E. hellem*, *E. cuniculi*, and *E. intestinalis* [10].

These microorganisms are widely speeded, but misunderstood, though they have a high importance for human health and agriculture. The medicinal importance of human pathogenic microsporidia increased when it was found to be responsible for clinical symptoms up to death in patients with suppressed immunity, AIDS patients, patients after organ transplants and oncological patients [22,23,24,25,26]. 

Microsporidia also affect agriculture due to their occurrence in animals farmed for human consumption (ruminants, pigs, fish, etc.); [27,28,29,30] companion animals; pets [31,32,33]; and their occurrence in wild animals and birds [34,35,36,37].

## 2. Microscopic, Serological and Immunological Diagnosis of *Encephalitozoon* spp. and *Enterocytozoon bieneusi*

The diagnostic of microsporidiosis is ordered mostly in patients, although non-specific symptoms occur, depending on the host species. The probability of microsporidian infection is, of course, higher in immunodeficient patients, not only in HIV patients and animals suffering from viral infections targeting the immune system, but also in recipients of organ transplants, post-chemotherapy patients and so on. In these patients, we should approach the diagnostics of microsporidia in cases of prolonged bloodless diarrhea (humans, suis, younglings, carnivores, rodents), which can be accompanied with severe weight loss. In rabbits and hares we proceed to diagnostics of microsporidiosis after the occurrence of neurological symptoms, such as torticollis and limb weakness. In immunocompetent patients, the possibility of finding microsporidia is severely lower, although they should be examined for the presence of microsporidia in cases of watery diarrhea, where no other causative agent was found in laboratory tests, or when usual therapy (for example with antibiotics) has no effect, in cases of dysuria or other nephrological alterations (carnivores), or in cases of neurological disorders (rabbit, carnivore cubs). Microsporidian infection in immunocompetent patients can end as a self-limiting disease after mild symptoms appear, similar to cases of cryptosporidiosis and isosporosis.

The selection of laboratory diagnostic methods in the diagnostics of microsporidiosis is dependent on the technical equipment of the laboratory, mastered diagnostic methods and on the professional abilities and expertise of the personnel. Diagnostics of microsporidian infections is quite complex, multistage and prolonged, mostly because of the difficult resolution against distinction from bacteria and small yeast in clinical samples. This means that more than one method is needed to confirm the diagnosis. As a screening method, it is suitable to examine feces, urine, sputum and other samples with the use of optical brightening agents (if the laboratory owns a fluorescent microscope [10]) or with chromotrope staining by Weber [38], enabling the ruling out of negative samples. 

With the help of optical brighteners, the spores are visualized in a fluorescent microscope due to the binding of the optical brighteners to the chitin in the spore wall. Depending on the reagent used as well as the wavelength, the walls of the microsporidium spore will fluoresce. Using Uvitex 2B, Rylux D (Ostacolor, Praque, Czech Republic) and a wavelength of 405–490 (light on observation, 510 nm), the spores will appear as green-white or turquoise oval formations (Figure 1). 

In Weber trichrome staining, the spore wall is stained pink-red, with a bright interior of the spore, or possibly with horizontal or diagonal stripes, which represents the polar tube. The background appears green or green-yellow.

Another alternative is the use of commercially manufactured monoclonal antibodies for examination of feces for the presence of *Encephalitozoon intestinalis* and *Enterocytozoon bieneusi*, diagnosed in humans with AIDS [24,39]. However, in certain stages of infection (early stage of primoinfection, when spores are not excluded until six weeks) and in cases of chronic infections (mostly in rabbits, when spore exclusion is postponed but clinical signs still remain), the exclusion of spores via feces is not the rule. This is why it is needed to complete the diagnostics with a serological IFAT method when the number of samples is small (up to 10 samples) or with ELISA in case of larger amounts of samples to detect antibodies against clinically significant species of microsporidia. *Encephalitozoon* spp. spores are obtained from infected cell lines, e.g., VERO E6, RK13 and MDBK [40,41]. 

If the presence of microsporidia is confirmed, determination of the species causing the infection is needed. This is made possible mainly by transmission electron microscopy. Another method is the concentration of spores by sedimentation, DNA isolation and species determination with the use of molecular methods. Molecular methods are advancing rapidly and they allow easy and immediate diagnostics of microsporidia in feces, urine and sputum with the possibility of excluding previous steps [22,36].

## 3. Sample Processing and DNA Extraction

DNA extraction from microsporidian spores is very difficult due to the presence of chitin in the cell wall. Its presence requires creating proper conditions, in which the disruption of the chitin cell wall will occur. The most commonly used methods are: mechanical disruption with the use of glass and zircon beads combined with proteinase K [36,42], ultrasound or the method of freezing in nitrogen and subsequent defrosting [43]. Some authors recommend incubation with enzymes disrupting chitin (chitinase, lyticase), during which the DNA is released from the spore [44,45].

Microsporidian DNA can be isolated from clinical samples such as tissues, swaps from cornea and mucosa, duodenal aspirates, samples of urine and feces, but also from in vitro cultures using commercial DNA extraction kits (AmpliSense, QIAGEN, etc.), or by routine methods such as proteinase K digestion protocol, followed by phenol-chloroform extraction and ethanol precipitation [46]. DNA is also isolated from samples conserved in paraffin using standard methods [47], or with commercial kits, such as DexPAT (Takera Biochemical, Berkeley, CA, USA). Successful isolations of DNA from microscope slides (slides with Giemsa and trichrome staining) [48], in which scrapes were used as material for isolation were observed. In the case of feces samples, some authors recommend purifying samples with 0.5% sodium hypochlorite, 10% formalin or 1 M KOH [49,50], while other authors suggest only dilution of the samples [51]. The extraction using FTA filters, which are saturated with denaturation agents, chelating agents and free radical traps is very effective. FTA filter paper and the QIAamp stool mini kit could detect *E. bieneusi* at concentrations as low as 800 spores/mL [52]. The method used for DNA isolation can significantly influence the sensibility of the reaction, therefore the choice needs to be made carefully.

## 4. Diagnostic Methods Based on Nucleic Acid Analysis

In comparison to the traditional methods, molecular methods can grant certain benefits, mostly higher sensitivity, higher accuracy and simplicity. They are based on working with nucleic acids and synthetically manufactured or cloned molecules of DNA, which are specific to the DNA sequences of the pathogen (primers, probes). The most commonly used molecular method is the polymerase chain reaction (PCR), in which a pair of primers is annealed on the template DNA and this DNA is subsequently copied again and again, in the presence of free nucleotides and polymerase enzymes [53]. The amplification of the target DNA of the pathogen (i.e., amplicon, which is the final product of PCR) provides two advantages: better sensitivity in comparison with the use of probes, and possible subsequent analyses of the amplicon (e.g., restriction analyses, sequencing).

Until now, large numbers of methods based on PCR were published for the amplifications of SSU and the LSU region of rRNA genes, and also for the intergenic regions (ITS, Figure 2) for the diagnostic and species differentiation of microsporidian species infecting humans and animals.

In the GeneBank database, SSU rRNA gene sequences of several microsporidia pathogenics for mammals are available: *Encephalitozoon cuniculi*, *Encephalitozoon hellem*, *Encephalitozoon intestinalis*, *Enterocytozoon bieneusi*, *Vittaforma corneae* etc. [54]. Moreover, PCR primers for SSU rRNA for the identification of microsporidia in clinical samples were designed. With this method, it is possible to identify microsporidia on a species level without the need of the ultrastructural examination.

For the amplification of microsporidia from clinical samples, common methods of tissue biopsy, corneal scrapes, cultivated microorganisms and samples of urine are sufficient. The necessary conditions in the diagnostics of microsporidia are adequate laboratory facilities and a good and workmanlike manner.

The first sequence readouts from SSU rRNA of microsporidia were published by Vossbrinck et al. in 1987 [55] (accessory number M24612). Subsequently, PCR with primers amplifying preserved SSU gene regions of *V. necatrix* for the amplification and DNA fragments sequencing of the ribosome small subunit (SSU) rRNA genes and other mammalian microsporidia was used [56,57,58,59]. Today, several nucleotide sequences of the SSU and LSU rRNA gene are known in different species of microsporidia, including genus *Encephalitozoon* (*E. cuniculi*, *E. hellem*, *E. intestinalis*) and *Enterocytozoon* (*E. bieneusi*). All are accessible in GenBank and EMBL databases. The first described PCR diagnostic of microsporidiosis was done by Zhu et al. [58], using primer pairs and hybridization probes for *E. bieneusi*. In the same year, Zhu et al. [57] published nucleotide sequences of the SSU rRNA of *Septata intestinalis (E. intestinalis).* The primer pair V1 and EB450-amplified cloned regions of the SSU rRNA of *E. bieneusi* and DNA from tissues infected with *E. bieneusi*. Weiss et al. [60] amplified cloned regions of SSU rRNA of *E. intestinalis* and DNA from tissues, body fluids and feces infected with *E. intestinalis* and cell cultures with the use of primer pairs V1 and SI500 and hybridization probes for *E. intestinalis*. Visvesvara et al. [56,61] designed two primer pairs for species-specific amplification of DNA fragments of *E. cuniculi* (positions 344–364 and 872–892 of the SSU rRNA region of *E. cuniculi* (ECUN-F and ECUN-R)) and for *E. hellem* (positions 358–378 and 884–904 of the SSU rRNA region of *E. hellem* (EHEL-F and EHEL-R)). These two primer pairs were used for the species differentiation from cultivated organisms and organisms found in different clinical samples [43,62]. Katzwinkel-Wladarsch et al. [49,63] described a nested PCR assay, enabling the diagnostic of four species of microsporidia (*E. bieneusi*, *E. hellem*, *E. cuniculi* a *E. intestinalis)* with the concentration of 3–100 spores in 0.1 g of fecal sample, as well as in different samples, such as sputum, urine, cerebrospinal liquor and other.

Species non-specific primer pairs. Species non-specific primer pairs in selected microsporidian species are summarized in Table 1. Primer pair PMP1 (nucleotide sequence of this primer is identical with primer V1)/PMP2 amplifying only the SSU rRNA region is capable of detecting five microsporidian species (*E. bieneusi*, *E. cuniculi*, *E. intestinalis*, *E. hellem* and *E. bieneusi*, Table 1). This nonspecific primer pair is currently the most widely used in the examination of clinical samples [29,64,65]. For the confirmation of the amplicon identity, restriction enzymes *Pst*I and *Hae*III are used. *E. bieneusi* does not have a restriction site for *Pst*I, only the *Encephalitozoon* spp. amplicon is cleaved in two fragments [66]. A similar principle applies in primer pair C1 (part of V1 primer sequence)/C2, identifying four microsporidian species (*E. bieneusi*, *E. cuniculi*, *E. intestinalis* and *E. hellem*) and restriction enzymes *Hin*dIII and *Hin*fI, which can determine their identity. The amplicon of *E. bieneusi* shows one restriction site for *Hin*dIII and for the identification of *Encephalitozoon* spp., the second enzyme *Hin*fI is used, with a different number of restriction sites (one for *E. cuniculi*, two for *E. hellem* and three for *E. intestinalis*). Primer pair V1/1 492 is almost identical, and amplifies a large section of the SSU rRNA gene of four microsporidian species (*E. bieneusi*, *E. cuniculi*, *E. intestinalis* and *E. hellem*). 

Another primer pair identifying species from genus *Encephalitozoon* and other microsporidian species is int530f/int580r. It amplifies a product including a large segment of the SSU rRNA gene, ITS region and a small segment of the LSU rRNA. Species differentiation can be done by Southern blot, with the use of chemiluminescent species-specific hybridization probes for species *E. cuniculi*, *E. hellem* and *E. intestinalis*, or by DNA sequencing [66].

Besides classic PCR methods with one primer pair, nested PCR tests were described, using two primer pairs. The first primer pair MSP1/MSP2A amplifies a large region containing ITS, SSU and LSU rRNA genes of several species (*E. hellem*, *E. cuniculi*, *E. intestinalis*, *V. necatrix*, *V. lymantriae*, *Ameson* (*Nosema*) *michaelis* and *Ichthyosporidium giganteum*), for which a unique “downstream” primer for *E. bieneusi* was designed. For the second PCR, primer pairs were MSP3/MSP4B identifying *E. bieneusi* and MSP3/MSP4A for *E. cuniculi*, *E. hellem* and *E. intestinalis*. *E. bieneusi* was differentiated from *Encephalitozoon* spp. due to a different length of the amplicon, and *E. cuniculi* from *E. intestinalis* using a restriction endonuclease *Mnl*I, creating a restriction fragment of different lengths (*E. cuniculi*—289 bp and *E. intestinalis*—305 bp; Figure 3) [49,65].

Another nested PCR test detects only four microsporidian species (*E. bieneusi*, *E. hellem*, *E. cuniculi* and *E. intestinalis*). In the first reaction, with the use of an upstream primer Mic3U and a downstream primer Mic421U, a fragment of a length of 410–433 bp is amplified and with the second PCR with the upstream primer Mic266 and downstream primers (Eb379, Ec378, Eh410 and Ei395), individual species are determined [68]. Mirjalali et al. [69] developed and successfully implemented two pairs of primers amplifying the SSU rRNA gene for nested PCR. The primers of the first pair PMicF and PMicR form a fragment of the 779 bp fragment SSU rRNA of *Encephalitozoon* spp. and *E. bieneusi*. In the second nested PCR reaction, EnbF and EnbR primers were used to amplify 440 bp for *E. bieneusi*, and EncepF and Aceptac primers to amplify 629 bp fragments to amplify *Encephalitozoon* spp. 

At our department, we have designed a primer for genus *Encephalitozoon* (*E. cuniculi*, *E. hellem* and *E. intestinalis*), which is a genus-specific primer amplifying the whole ITS region, small regions of the SSU rRNA and LSU rRNA genes with an annealing temperature of 60 °C (Figure 3). Forward primer ecfITSf is situated on sequence positions from 2920 to 2938 in *E. cuniculi* (AJ005581.1), from 1187 to 1205 in *E. hellem* (AF272836.1) and from 2644 to 2627 in *E. intestinalis* (CP001951.1) in the GenBank database. Reverse primer ecfITSr is situated on the positions of the same sequences, from 3252 to 3270 in *E. cuniculi*, from,1528 to 1546 in *E. hellem* and from 2327 to 2310 in *E. intestinalis* [35].

Species-specific primer pairs. Species-specific primer pairs in selected microsporidian species—*E. bieneusi*, *E. cuniculi*, *E. hellem* and *E. intestinalis*—are summarized in Table 2. For species determination of *E. bieneusi*, five primer pairs were designed until now (EBIEF1/EBIER1, V1/EB450, V1/Mic3, Eb.gc/Eb.gt and EbF/EbR). In three of them, the specificity of amplicons was confirmed either by DNA sequencing, Southern blot or with restriction enzymes (V1/EB450, V1/Mic3 and EbF/EbR). The primer pair V1/EB450 amplifies the SSU rRNA gene region and creates an amplicon of 348 bp in length, and the hybridization probe EB150 confirms its identity. With primers EbF/EbR, an enormously long amplicon (up to 1265 bp) from the SSU rRNA gene, and a hybridization probe is also used. Restriction endonucleases *Mnl*I and *Dde*I will confirm the identity when using primer pair V1/Mic3. Primer pair Eb.gc/Eb.gt is the only one that also includes the ITS and LSU rRNA regions of the gene, but the amplicon has the shortest length—only 210 bp.

Four primer pairs were described for *E. intestinalis*, amplifying only the SSU rRNA gene region (V1/Si500, 3/3, SINTF1/SINTR and V1/Sep1). Hybridization probes were designed only for primer pairs V1/Si500 (SI60) and 3/3 (Table 2).

There is a different situation in *E. cuniculi* and *E. hellem*, in which primer pairs amplifying the SSU rRNA gene for specific identification were designed (*E. cuniculi*: ECUNF/ECUNR and *E. hellem*: EHELF/EHELR).

The PCR method is also useful for the identification of unknown microsporidian species in human, but also in animal, infections. The use of phylogenetically conserved primers (amplifying the ITS region, SSU and LSU rRNA genes) enables cloning and subsequent sequencing of rRNA gene fragments of yet uncharacterized microsporidia (Table 3) [78]. These rRNA sequence readouts can be used in phylogenetic analyses using a BLAST program or other similar programs comparing unknown sequences of rRNA with the sequences of know Microsporidia, which are available in the GenBank database.

Primer pairs V1(18f)/1492r and 530f/580r are considered “universal” in terms of their ability to effectively amplify unknown rRNA genes of new species or genotypes of microsporidia [79].

Some authors have published diagnostic methods for microsporidia with the use of real-time PCR, which has the advantage of quantitative valuation of the sample [84,85]. Real-time PCR can detect the accumulation of amplicons in real time by a number of technologies. In microsporidia, diagnostic methods were described either with the use of fluorescent dyes (SYBRGreen) intercalating directly to DNA [29,35,36], or with the use of a fluorescent probe (TaqMan). Real-time PCR is commonly used in the “multiwell” format, from which no further post-amplification sample processing is needed, decreasing the risk of contamination associated with PCR procedures.

In 2003, Menotti et al. [84,86] focused on the design of probes and primers amplifying the SSU rRNA gene in *E. bieneusi* and *E. intestinalis* (FEB1/REB1 a FEI1/REI1). The reference value of the detection of spores was assessed to 20 spores per milliliter. A multiplex real-time PCR [85]), which simultaneously detects *E. bieneusi*, *E. cuniculi*, *E. hellem* and *E. intestinalis* in fresh and in formalin conserved samples with the use of probes (*EbITS-114revT* and *Eint82Trev*) and primers detecting the SSU rRNA gene region (EbITS-89F/EbITS-191R and MSP1F/Eint227R), was also published.

Fluorogenic-5‘nuclease PCR tests are fitting for the quick and sensitive detection of *E. hellem*, *E. cuniculi* and *E. intestinalis.* Hester et al. [87] designed a fluorescent probe (*EncephP1*) and species-specific primers for *E. cuniculi*, *E. hellem* and *E. intestinalis* (EcunF1/EcunR2, EhelF1/EhelR2 and EintF1/EintR2), amplifying the SSU rRNA gene. Although their method was verified only on non-contaminated samples containing microsporidian DNA, its use in clinical samples is also expected.

## 5. MLST (Multilocus Sequence Typing)

Current genotyping of *E. bieneusi* is based only on molecular methods, because it is not possible to morphologically distinguish the genotypes. Sequencing the ITS regions of rRNA genes is a standardized method in genotyping, due to the high variability in this region between individual genotypes. Based on results from 2009, a new classification of *E. bieneusi* genotypes was proposed [88]. Most primers published from different authors amplify the ITS region for genotyping *E. bieneusi*, which consists of 243 base pairs. Some sequences of the mentioned genotypes include also the SSU and LSU rRNA regions of genes. Different terminologies, used by different authors for the description of *E. bieneusi* genotypes, created difficulties in the identification of not only known, but also newly discovered, genotypes. The absence of suitable nomenclature in the description of genotypes has consequences in numerous designations of one genotype; therefore some genotypes have several designations. For example, genotype D has six designations (PigITS9 (AF348477), WL8 (AY237216), Peru9 (AY371284), PtEb VI(DQ885582), and CEbC (EF139197)). This vast genetic diversity in *E. bieneusi* as a species (not only from differences in the ITS region) has contributed to uncertainties in the role of *E. bieneusi* as an infectious agent, because it is difficult to assess how different genotypes affect the host.

Lately we have noticed genetic population studies describing the combined use of four microsatellite and minisatellite markers sufficient for distinguishing *E. bieneusi*. This is known as multi-locus sequence typing (MLST; [89]). This method was successfully used for re-examination of samples containing *E. bieneusi* from patients suffering from AIDS in Peru [90], samples from zoo animals and samples from equine, calf and cattle husbandry in China [91,92,93,94]. However, these observations of *E. bieneusi* genotypes will have to be proven in other geographical sites. To this day, more than 240 genotypes were identified in various animal hosts [95,96]. With the help of phylogenetic analysis, ITS genotypes of *E. bieneusi* were divided into nine different clusters (http://dx.doi.org/10.1002/9781118395264.ch3 [97]. The first cluster (group 1) contains 94% of the published *E. bieneusi* genotypes, and it was established to have a zoonotic potential [91]. In comparison, genotypes in the other eight clusters (groups 2 to 9) are present mainly in specific hosts and in wastewaters [98,99]. For better understanding of the taxonomy and molecular characteristics of *E. bieneusi*, a multilocus sequence designation has been developed (MLST), with the use of three microsatellites (MS1, MS3 and MS7) and one minisatelite (MS4) as markers Table 4; [89,100].

In recent years, the use of microsatellite and minisatellite genetic markers in genetic population studies has improved our understanding of the transmission of parasitic protozoa [101,102]. It was proved that some parasitic protozoa with sexual reproduction in their life cycle can have a structure as a clonal population, such as *Toxoplasma gondii* and *Cryptosporidium hominis.* However, population structures of parasites (e.g., *Plasmodium falciparum* and *Trypanosoma brucei*), can differ in effect of various factors, such as hosts, host migration, geographic locations and transmission intensity [101]. The assessment of the genetic structure of the population is vital for understanding transmission routes. Because the specific transmission routes of *E. bieneusi* remain unknown and there are no effective medications for the complete therapy of *E. bieneusi* infection in humans or animals, scientists, doctors or managers of agricultural corporations should be given notice to take precautions for controlling the contamination of the environment with these parasites.

## 6. Diagnostic Procedures—Recommendation

As a screening, we recommend that the first use of examination of feces, urine, sputum, and feces is most often used, which must be subjected to ether extraction before staining. Dilute approximately 1 g of feces in 6 mL of water, pour over the gauze, add 6 mL of diethyl ether and a stopper, mix thoroughly and centrifuge for 2 min. at 600× *g*. Release the ring at the ether–water interface and discard the supernatant. The sediment can be used for culture, for DNA isolation, or after fixation on a staining slide. In our laboratory, staining with optical brighteners (RyluxD, Calcofluor) has proven successful, with the help of which we can exclude negative samples. It is very advantageous to use a positive control (e.g., stored *Encephalitozoon* spp. spores from cell culture or older positive samples) to compare the size and shade of fluorescence. Spores of *E. bieneusi* are easily overlooked and it is good to realize that they are even smaller than *Encephalitozoon* spp. For human samples, another option is to use a commercially produced monoclonal antibody assay to examine feces for the presence of *E. intestinalis* and *E. bieneuses*, which are most commonly diagnosed in people with AIDS. However, in some stages of infection (at the beginning of primary infection, when spore excretion does not begin until 6 weeks), in chronic infection (when spore excretion is stopped but neural clinical signs persist), excretion of spores by excrement may not be the rule. Therefore, microscopic examination should be supplemented by serological IFAT testing on a small number of samples (up to 10) or ELISA on a larger number of examined samples, for the presence of antibodies against clinically important species of microsporidia.

If the presence of microsporidia is detected, the species of the agent still needs to be identified. Sediments concentrated by sedimentation are used for DNA isolation and determination by molecular methods. Prior to DNA isolation, the sample must be pre-isolated. At our workplace, we use a Precellys 24 sample homogenizer (Bertin Technologies, GmbH, Frankfurt am Main, Germany), where we break the samples with glass (0.5 mm) and zirconium beads (1.0 mm) and centrifuge at 6500 rpm for 90 s in 300 μL of lysis solution. We then isolate the DNA using commercial DNA extraction kits (QIAGEN or AmpliSense, Moskva, Russia). The method used to isolate DNA can significantly affect the sensitivity of the reaction, so the choice must be done carefully. In our practice, an effective method for the molecular detection of samples infected with microsporidia involves the use of a classical PCR reaction using universal primers to amplify the SSU rRNA gene regions of four human and animal infectious pathogen microorganisms (*E. bieneusi*, *E. intestinalis*, *E. cuniculi* and *E. hellem*), which are PMP1 and PMP2. After sequencing and identification of *E. bieneusi*, positive samples are used to amplify the ITS region for genotyping using a specific primer pair MSP3 and reverse MSP4B. If they are *Encephalitozoon* spp. a species-specific primer pair per ITS region is used (int530f and int580r). The second PCR method we use is SYBR Green Real-Time PCR using the primer pairs ecfITSf and ecfITSr we created for the primers *E. cuniculi*, *E. intestinalis* and *E. hellem*, followed by DNA sequencing and genotyping. For *E. bieneusi* we use the primer pair MSP3 and reverse MSP4B as in classical PCR. Quick diagnosis in clinical samples without genotyping uses primers ECUNF, ECUNR, SINTF, SINTR, EHELF, EHELR, EBIENF, EBIENR.

## 7. Conclusions

Microsporidia are one of the most common and most species-numerous parasites in animals. We encounter them unknowingly on a daily basis, often, we even host them, yet, by the virtue of their hidden way of life, general awareness of microsporidia is very low. Light-microscopic visualization of the spores can be deemed as a standard technique for diagnosis of these intestinal parasites, however, due to the limited sensitivity and unfeasibility of species differentiation with this staining method, PCR has become a better alternative. PCR is thus a desirable method for confirmation and identification of microsporidia thanks to its high sensitivity, speed and specificity.

## Figures and Tables

**Figure 1 jof-06-00114-f001:**
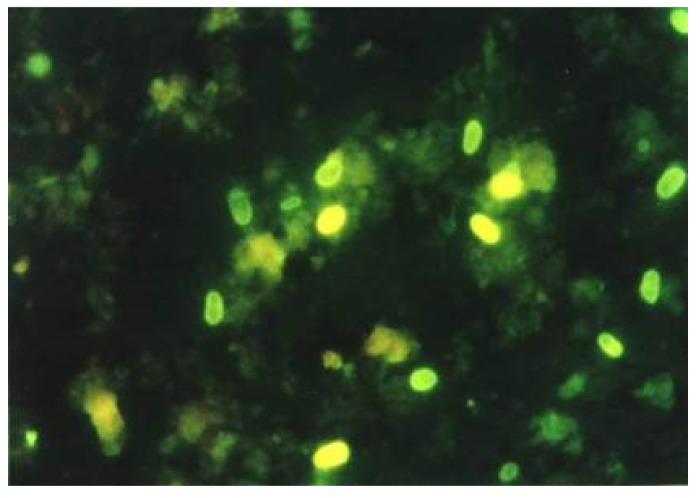
Sample stained with Rylux-green fluorescence (original photo, 1000×).

**Figure 2 jof-06-00114-f002:**
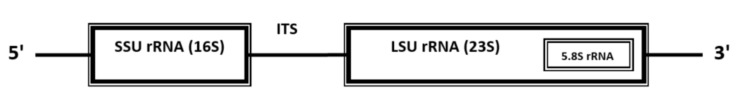
Illustration of rRNA genes and ITS (internal transcribed spacer) region in Microsporidia 5.8S rRNA gene is fused in the LSU rRNA gene.

**Figure 3 jof-06-00114-f003:**
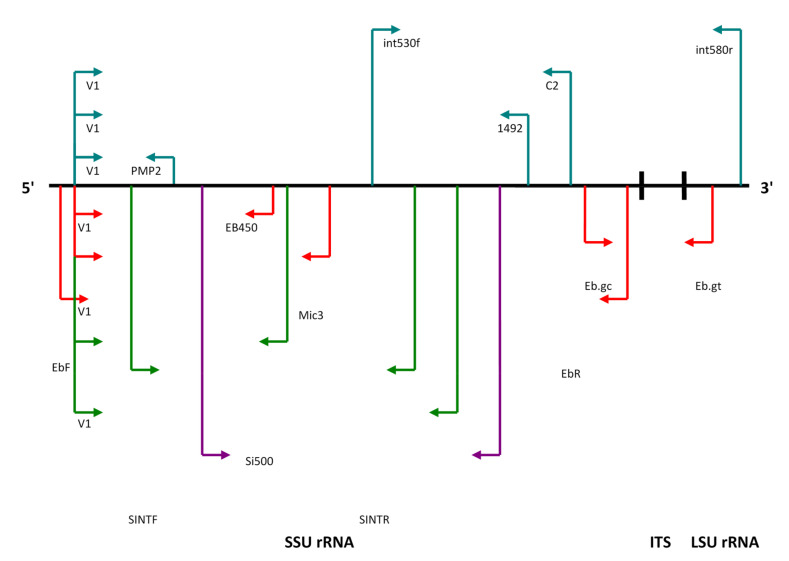
Diagram of ITS areas and rRNA genes containing positions of species non-specific and specific primer pairs *Enterocytozoon bieneusi*, *Encephalitozoon cuniculi*, *Encephalitozoon hellem* and *Encephalitozoon intestinalis*. Source: [61]. Blue color—species non-specific primer pairs. Red color—primer pairs for *E. bieneusi*. Green color—primer pairs for *E. intestinalis*. Violet color—primer pairs for *E. cuniculi* and *E. hellem*.

**Table 1 jof-06-00114-t001:** Species non-specific primer pairs in selected *Encephalitozoon* spp. and *Enterocytozoon bieneusi*.

Species	Sequence (5′→3′)	Primer Designation	T_a_	bp	Ref.
*Encephalitozoon* spp. and *E. bieneusi*	-CACCAGGTTGATTCTGCCTGAC- -CCTCTCCGGAACCAAACCCTG-	PMP1(V1) PMP2	60	Eb 250 Ec 268 Ei 270 Eh 279	[67]
*Encephalitozoon* spp. and *E. bieneusi*	-CACCAGGTTGATTCTGCC- -GTGACGGGCGGTGTCTAC-	C1(part V1) C2	56	Eb 1 170 Ec 1 190 Eh 1 205 Ei 1 186	[45]
*Encephalitozoon* spp. and *E. bieneusi*	-CACCAGGTTGATTCTGCCTGAC- -GGTTACCTTGTTACGACTT-	V1 1 492	-	around 1200	-
*Encephalitozoon* spp.	-TGCAGTTAAAATGTCCGTAGT- -TTTCACTCGCCGCTACTCAG-	int530f int580r	40	around 1000	[42]
Nested PCR *Encephalitozoon* spp. and *E. bieneusi*	-TGAATG(G/T)GTCCCTGT- -TCACTCGCCGCTACT-	MSP1–PR MSP2A	58	-	[49]
	-GGAATTCACACCGCCCGTC(A/G)(C/T)TAT- -CCAAGCTTATGCTTAAGT(C/T)(A/C)AA(A/G)GGGT-	MSP3–DR MSP4A	58	Ec 289 Ei 305	
	-TGAATG(G/T)GTCCCTGT- -GTTCATCGCACTACT-	MSP1–PR MSP2B	58	−	
	-GGAATTCACACCGCCCGTC(A/G)(C/T)TAT- -CCAAGCTTATGCTTAAGTCCAGGGAG-	MSP3–DR MSP4B	58	Eb 508	
Nested PCR *Encephalitozoon* spp. and *E. bieneusi*	-CCAGGUTGATUCTGCCUGACG- -TUACCGCGGCUGCUGGCAC-	Mic3U–PR Mic421U	65	Eb 410 Ec 419 Ei 421 Eh 433	[68]
	-AAGGAGCCTGAGAGATGGCT- -CAATTGCTTCACCCTAAGGTC- -GACCCCTTTGCACTCGCACAC- -TGCCCTCCAGTAAATCACAAC- -CCTCCAATCAATCTCGACTC-	Mic266–DR Eb379 Ec378 Eh410 Ei395	62	Eb 132 Ec 113 Eh 134 Ei 128	
Nested PCR	-GGTTGATTCTGCCTGACG- -CTTGCGAGC(G/A)TACTATCC-	PMicF PMicR	55	Eb 779 Enc 779	[69]
	-GGTAATTTGGTCTCTGTGTGTGTG-	EnbF	57	Eb 440	
	-CTACACTCCCTATCCGTTC-	EnbR			
	-AGTACGATGATTTGGTTG- -ATCCACAT-ACCA-	EncepF Aceptac		Enc 629	
Real-Time PCR *Encephalitozoon* spp.	-TGTACACACCGCCCGTCG- -TTTCACTCGCCGCTACTC-	ecfITSf ecfITSr	60	Ec 349 Ei 334 Eh 359	[35]

T_a_—primer/template annealing temperature; bp—length of amplicon in base pairs; Eb—Enterocytozoon bieneusi; Ec—Encephalitozoon cuniculi; Ei—Encephalitozoon intestinalis; Eh—Encephalitozoon hellem; Enc—Encephalitozoon.

**Table 2 jof-06-00114-t002:** Species-specific primer pairs in *Encephalitozoon* spp. and *Enterocytozoon* sp.

Species	Sequence (5′→3′)	Primer Designation	T_a_	bp	Reference
*E. bieneusi*	-GAAACTTGTCCACTCCTTACG- -CCATGCACCACTCCTGCCATT-	EBIEF1 EBIER1	55	607	[65]
*E. bieneusi*	-CACCAGGTTGATTCTGCCTGAC- ACTCAGGTGTTATACTCACGTC-	V1 EB450	48	353	[58,70]
*E. bieneusi–*RE1	-CACCAGGTTGATTCTGCCTGAC- -CAGCATCCACCATAGACAC-	V1 Mic3	54	446	[71,72]
*E. bieneusi*	-TCAGTTTTGGGTGTGGTATCGG- -GCTACCCATACACACATCATTC-	Eb.gc Eb.gt	49	210	[73]
*E. bieneusi*	-GCCTGACGTAGATGCTAGTC- -ATGGTTCTCCAACTGAAACC-	EbF EbR	55	1294	[74]
*E. intestinalis*	-CACCAGGTTGATTCTGCCTGAC- -CTCGCTCCTTTACACTCGAA-	V1 Si500	58	375	[60]
*E. intestinalis*	-GGGGGTAGGAGTGTTTTTG- -CAGCAGGCTCCCTCGCCATC-	3 3	65	930	[75]
*E. intestinalis*	-TTTCGAGTGTAAGGAGTCGA- -CCGTCCTCGTTCTCCTGCCCG-	SINTF1 SINTR	55	520	[76,77]
*E. intestinalis*	-CACCAGGTTGATTCTGCCTGAC- -CCTGCCCGCTTCAGAACC-	V1 Sep1	55	824	-
*E. cuniculi*	-ATGAGAAGTGATGTGTGTGCG- -TGCCATGCACTCACAGGCATC-	ECUNF ECUNR	55	549	[56,77]
*E. hellem*	-TGAGAAGTAAGATGTTTAGCA- -GTAAAAAGACTCTCACACTCA-	EHELF EHELR	55	547	[56]

T_a_—primer/template annealing temperature; bp—length of amplicon in base pairs; RE1—possibility of the use of restriction enzymes MnII and DdeI for confirmation of amplicon identity (RFLP).

**Table 3 jof-06-00114-t003:** Primer pairs amplifying rRNA genes of new species or genotypes of *Encephalitozoon* spp. and *Enterocytozoon* sp.

Species	Sequence (5′→3′)	Target Region	Primer Designation	bp	Reference
*E. bieneusi*, *E. hellem*, *E. intestinalis*, *E. cuniculi*, other species	-ACCAGGTTGATTCTGCCTGAC- -GGTTACCTTGTTACGACTT-	SSU rRNA	V1 1492	1200	[57,58,59,60]
*E. bieneusi*, *E. hellem*, *E. intestinalis*, *E. cuniculi*, other species	-CACCAGGTTGATTCTGCC- -TTATGATCCTGCTAATGGTTC-	SSU rRN	18f 1537r	1300	[80]
*E. intestinalis*, *E. cuniculi*, *E. hellem*, other species	-CACCAGGTTGATTCTGCCTGA- -TTATGATCCTGCTAATGGTTCTCCAAC-	SSU rRNA	Micro-F Micro-R	1300	[56,81]
*E. bieneusi*	-CACCAGGTTGATTCTGCCTGA- -CAACTGAAACCTTGTTACGACTT-	SSU rRNA	Micro-F 1492N4	1200	[81]
*E. bieneusi*, *E. hellem*, *E. intestinalis*, *E. cuniculi*, other species	-CACCAGGTTGATTCTGCCTGACG- -TTATGATCCTGCTAATGGTTCTCC-	SSU rRNA	− −	1300	[75]
*E. bieneusi*, *E. hellem*, *E. intestinalis*, *E. cuniculi*, *V. cornae*, other species	-GTGCCAGC(C/A)GCCGCGG- -GGTCCGTGTTTCAAGACGG-	SSU rRNA, ITS, LSU rRNA	530f 580r	1550 1350 1350 1350 1300	[59,60,82,83]
*E. intestinalis*, *E. hellem*, *E. cuniculi*, other species	-TGCAGTTAAAATGTCCGTAGT- -TTTCACTCGCCGCTACTC-	SSU rRNA, ITS, LSU rRNA	int530f int580r	1000 1000 1000	[75,82]

**Table 4 jof-06-00114-t004:** Primer sequence of microsatellite and minisatellite loci of *Enterocytozoon bieneusi* for MLST analysis.

Locus	Sequence (5′→3′) Primers	Targeted Repeat	Annea-Ling Temp (°C)	bp	GenBank Accession No. (Nucleotide Positions)
MS-1	F1, CAAGTTGCAAGTTCAGTGTTTGAA R1, GATGAATATGCATCCATTGATGTT F2, TTGTAAATCGACCAAATGTGCTAT R2, GGACATAAACCACTAATTAATGTAAC	(TAT)_31_ (TAG)_11_	58 58	843 676	ABGB01000003 (63854–64529)
MS-2	F1, GTACAAGATGAAGTTCCTGAGT R1, CATGACATCATTTTACATACACAT F2, GGCCTGATAATAGATCGGATT R2, CAGCATCATCACACGTTCTCA	(TG)_19_	55 55	584 421	ABGB01001554 (367–787)
MS-3	F1, CAAGCACTGTGGTTACTGTT R1, AAGTTAGGGCATTTAATAAAATTA F2, GTTCAAGTAATTGATACCAGTCT R2, CTCATTGAATCTAAATGTGTATAA	(TA)_21_	55 55	702 537	ABGB01000035 (202–738)
MS-4	F1, GCATATCGTCTCATAGGAACA R1, GTTCATGGTTATTAATTCCAGAA F2, CGAAGTGTACTACATGTCTCT R2, GGACTTTAATAAGTTACCTATAGT	(TTATTTTTTCCATTTTT CTTCTTCTATTTCCTTTA)_9_	55 55	965 885	ABGB01000033 (1063–1947)
MS-5	F1, GTCATGATCACCGGCACTTA R1, CTCAAGGATCGTCAAGCTGA F2, GCAGGCTTTGCAGTTGGCTT R2, GTGAAGGAAGCCGTAGCTAA	(GCGGCTGGTTTCGCAG CAGCGGTTTTAGCAA CTGGCTTC)12	55 55	882 599	ABGB01000169 (546–1144)
MS-6	F1, GAATAGAATGATTCTAGCCATGA R1, CCATATAGCCTTTAAGACCAAA F2, CTTTTCAAGGATGGTTTGAATGA R2, CAAAGGGTACCTCCAATCAAA	(AT)_14_	55 55	706 498	ABGB01000562 (660–1157)
MS-7	F1, GTTGATCGTCCAGATGGAATT R1, GACTATCAGTATTACTGATTATAT F2, CAATAGTAAAGGAAGATGGTCA R2, CGTCGCTTTGTTTCATAATCTT	(TAA)_13_	55 55	684 471	ABGB01000014 (23807–24277)

Source: [89].

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
