# Peer review of "Alternatives in Molecular Diagnostics of Encephalitozoon and Enterocytozoon Infections"

_jof, 2020, doi:10.3390/jof6030114_

Round 1
Reviewer 1 Report
The authors review the molecular diagnostics of human pathogenic microsporidians. The intention to review this field is useful and would help to get an overview on the methods of clinical diagnostics and the different molecular approaches and primers. My main criticism at the present text is that it lacks any recommendation or discussion of the “gold standards” in this variety of primers and methods applied (or discussion that there is no gold standard because…). Furthermore, the structure of the text is not really satisfying.
Specifically:
A clear statement on the scope of the article is lacking. The title says “Diagnostics of microsporidian infections”, but apparently it is only about Encephalitozoon and Enterocytozoon.
Section 2 contains a lot of information on cases when diagnosis of microsporidians is recommended, but the actual description of the diagnostic techniques is only half of the section.
The differentiation of section 3 and 5 is not clear to me.
Most of the document consists of primer lists, but only few additional information is given. If I read this article, I would expect some suggestions what is suggested diagnostics. Normal PCR? Nested? Real-time? And which primers? The authors should use their expertise to give some guidance, which could make this paper a really valuable document.
Overall it is not very well written with sometimes strange wording and definitely needs language improvement. After considerable improvement, this manuscript could be published in JoF.
Author Response
Thank you very much for your comments. We accept most of them.
1. We change the title of the manuscript to "Alternatives in Molecular Diagnostics of Encephalitozoon and Enterocytozoon Infections"
2. We added some information to part 2 with the original photo.
3. We reworked part 3 and 5, respectively we canceled part 3 and inserted part of the text into part 5. We included part of the text in part 6 where we described our methodical procedures used in the diagnosis of these infections.
4. We had the manuscript checked by two English-speaking lecturers.
Yours sincerely
Reviewer 2 Report
The review entitled Alternatives in Molecular Diagnostics of Microsporidian Infections looks more deeply into the mechanisms of early detection of this type of infection. It places special emphasis on the combinatorial formulas of primers that serve to identify the different species of Enterocytozoon and Encephalitozoon.
The manuscript is well structured and the reasoning is easy to follow for readers not too close to this field of research. It seems to me that it can contribute to condense the current knowledge about the methods of identification of these species. Perhaps I have missed in the introduction a figure which includes the life cycle of these organisms and which helps to improve understanding and groups together the origin or sources of possible samples to be taken for subsequent identification by genetic or cellular methods.
On the other hand, authors should be careful when organizing the tables showing the oligonucleotides. I think they're totally disorganized. The data in each row should be well separated with enough space or with dividing lines. It is difficult to understand tables 1 and 2 in particular. Figure 2 should be also improved. A different organization, not only based on color codes, would allow a better understanding of which fragments are and a better coordination with the different tables.
A minor point is definition of an acronym. The definition of each acronym appears in a scattered manner many lines after it is first used.
The authors also have to check that all species names are italicized.
Author Response
Thank you very much for your comments. We tried to correct and supplement the required information.
1.We added a short development cycle to the first part.
2. We fixed the tables.
3. We checked the abbreviations and added their explanations.
4. We have corrected the names of species in italics.
Yours sincerely
Round 2
Reviewer 1 Report
The manuscript has been improved sufficiently to be published in JoF. I have only one recommendation:
Use the full title instead of the abbreviation in chapter 5, as it is not entirely clear what MLST means.